# Kind Regards in These Difficult Times: Anglo–Soviet Architectural Relations during the Second World War

**Ksenia Malich**

Design and Contemporary Arts Department, School of Design, National Research University Higher School of Economics, 101000 Moscow, Russia; kmalich@hse.ru

**Abstract:** The present article examines Anglo–Soviet architectural relations during the Second World War, the peculiarities of the perception of foreign experience, and the mutual professional interests. This paper aims to find evidence of multilateral and immensely diverse contacts and examine the reasons for and routes of such collaborations and the actors and institutions involved in the processes. This research attempts to construct new criteria for evaluating professional architectural relationships in the context of ideological and non-ideological obstacles. For this reason, this paper draws data from a wide range of sources, including the State Archive of the Russian Federation (GARF) and the Russian State Archive of Literature and Art (RGALI), the Schusev State Museum of Architecture (GNIMA), and research from the Royal Institute of British Architects (RIBA) Berthold Lubetkin and Erno Goldfinger archives.

**Keywords:** cultural diplomacy; British architecture; Soviet architecture; XX-century architecture





## 1. Introduction

For a long time, the issue of the Soviet Union's relations in architecture with Western European countries and Great Britain, in particular, was practically not studied because of ideological obstacles. It was slightly referenced in some papers dedicated to XX-century cultural policy. However, the reasons for and the results of this cooperation are intriguing and inspiring, and since the 1980s, there has been interest in this issue. It is essential to understand if any cooperation in architecture was possible under the circumstances of strained official relations. What kind of architectural influence could arise from this professional communication? To answer these questions, it is crucial to not only analyze the architectural practice itself but also to study the letters, discussions, and recollections of the architects themselves and other archival materials.

The specifics of the war period complicate the analysis of the issue, but at the same time, they expose the most pressing problems. Until now, communication between British and Soviet architects during the war has not been properly considered. Now, however, the historical distance allows us to work with documents and to build analysis in a more detailed and objective way. This research attempts to construct new criteria for evaluating professional architectural relationships in the context of ideological and non-ideological obstacles, considering the unprecedented emotional intensity and the scale of new city development problems. We will attempt to find out who the initiators of this cooperation were, what the final purposes of the professional contacts were, and if there were any differences between the official political agenda and the personal intentions of the architects.

## 2. Materials and Methods

As Stephen V. Ward relays in his research on the British planning movement's Soviet connections (one of the most essential papers dedicated to our subject), British intellectuals attracted to the Soviet Union were usually considered something between utopians or "quasi-religious" maximalists (Ward 2002; Ward 2012; Caute 1988; Hollander 1981). This

interpretation, which emerged during the Cold War and post-Cold War period, is strong indeed; however, it does not take into consideration the human factor when the role of professional contact went beyond conventional exchanges of "soft power" or "cultural diplomacy" or any governmental actors (Atkinson and Verity 2017, pp. 117–18; Gienow-Hecht 2010, p. 10). We also should indicate the difference between ideological and non-ideological and consider the fact that ideological prerequisites can be flexible and do not contradict the corporation. Quoting another author, Nigel Gould-Davies: "Was ideology important? . . . What was the relationship between ideology and interests?" (Gould-Davies 1999, p. 90).

But what should be the criteria for evaluating professional architectural relationships in the context of ideological and non-ideological obstacles? Let us here take into consideration three crucial moments. Firstly, the officially (on the levels of institutions and even governmental resolutions) declared intentions about the needs of the professional exchanges. Secondly, the way to implement such a policy. The third case is that we should analyze different types of documents concerning the professional interactions between Soviet and British architects. These include not only official documents (like correspondence between the institutions, reports in the press, and some direct orders) but also private letters and notebooks. And here, we may encounter another fourth circumstance: many personal documents, especially correspondence, still bear the imprint of self-censorship. So, it is not easy to develop a formal criterion for assessing this heritage, and sometimes we can only guess about the true intonations. However, we believe that a broader range of documents and new archive materials will lay the foundation for a more thoughtful understanding in the future and provide a more comprehensive and more objective view to highlight the true mutual interests hidden by the officially declared and constantly changing agendas.

In accordance with these tasks, I have divided the article into several paragraphs. The first section is dedicated to the official attempts to establish cultural ties with the help of VOKS. The following paragraph tells the story of the less-formal activities initiated by private organizations or even on personal levels during the years of the Anglo–Soviet Alliance. The following paragraph analyzes personal correspondence between British and Soviet architects during the war, and the last two sections attempt to understand the interests of architects of the Soviet Union in cooperating with British architects in general and in solving the problem of post-war restoration in particular.

This research incorporates a review of some academic publications on Soviet–British cultural relationships, but it mainly draws on unpublished materials from the State Archive of the Russian Federation (GARF) and the Russian State Archive of Literature and Art (RGALI), the Schusev State Museum of Architecture (GNIMA), and the Berthold Lubetkin and Erno Goldfinger archives from the Royal Institute of British Architects (RIBA). The study examines a series of documents of different types, including personal diaries, newspapers and magazines, official and private correspondence, brochures, advertising, and exhibition catalogs produced in the USSR and UK from the 1930s to the 1940s.

## 3. The First Attempts to Establish Architectural Ties

The intensive communication found during the Second World War was prepared during the interwar period. However, the situation of the 1930s was very different from the circumstances and reasons for professional correspondence in the first half of the 1940s. Great Britain was the first European country that formally recognized the USSR in 1924, and it was MacDonald who started to negotiate a treaty with the Soviet Union and established permanent diplomatic relations after a diplomatic break of 1927–1929.

The interwar period is considered the highlight of Soviet cultural diplomacy, but Anglo–Soviet relations were still marked by distrust. The Chairman of the VOKS Board (Vsesoiuznoe obshchestvo kul'turnykh sviazei s zagranitsei or the Society for Cultural Relations with Foreign Countries), Olga Kameneva, has outlined that the primary stake in this work should be placed on attracting the attention of the Western intellectuals since it was in that environment that public opinion was eventually formed. During the 1920s,

the communist utopia was based on confidence in the beneficial impact of the new social order, technological progress, and planned economy. That was what Europeans got the opportunity to see in the new Soviet world through the eyes of Herbert Wells, Bernard Shaw, and Walter Benjamin (Stern 2007). There were also other authors of numerous texts about life in Soviet Russia. However, since in the USSR, a special role was dedicated to literature, VOKS primarily tried to befriend writers. Secondly, it established contacts through cooperation with contemporary artists and the museum industry.

The Architectural Section, part of the VOKS art culture sector, intensified its work a little later. However, British architects became interested in coming to the USSR in the early 1930s through the initiatives of VOKS and "Intourist" tour operators (Kopp 1990; Bosma 2014; Malich 2020). Information about Soviet architecture was available in the form of reports, lectures, and brochures thanks to the work of the Society for Cultural Relations between the British Commonwealth and the USSR (Society for Cultural Relations with the USSR, or SCR, established in 1924). The Society received subsidies from VOKS but enjoyed absolute independence and freedom. It existed thanks to the private initiative of enthusiasts who were fond of Russian culture and sympathized with the Soviet social experiment.

The situation changed seriously after 1932, when "National Traditionalism" (socialist realism) was chosen as a more appropriate style for Soviet architecture. The censorial functions were carried out by the Union of Architects of the USSR and the Academy of Architecture. This control was based mainly on the results of the most important competitions, determined by the country's senior leaders, even by Stalin himself (Chmelnizki 2013, p. 323; Konysheva 2018). Against the background of the political situation, the atmosphere of distrust grew. After visiting London, Paris, and Stockholm in 1936, architect Alexander Gegello was preparing a speech for colleagues, noticing that the journey had opened his eyes to the "opposite of the two worlds of socialist and capitalist; the gulf that separates our union from the Western European countries."[1] Even this control was unofficial, presented as a professional discussion, though this conversation was getting tougher and tougher (Arkin 1936; Cohen 1987). At the same time, in 1937, the last departments of international architectural communications were closed (the Bureau of Scientific Communication with Abroad at the Academy of Architecture and the foreign section of the Union of Soviet Architects) (Konysheva 2021). Then all contacts went only through the VOKS architectural section. It became more difficult to organize events and SCR as well.

The next step was in 1943 when the Committee for Architectural Affairs (Komitet po delam arkhitektury) was set up under the Council of People's Commissars of the USSR to execute centralized control of architecture during wartime (Chmelnizki 2013, p. 323). This centralization caused the freezing of architectural relationships again. Before that, in 1941–1943, these contacts increased significantly. Nonetheless, in June 1941, Hitler broke the German–Soviet pact and invaded the Soviet Union.

## 4. Anglo–Soviet Alliance and the SCR Activities

The reports from the battle-fronts about the feat of the Russian resistance destroyed cities, and the siege of Leningrad (the costliest siege in history due to systematic starvation) shifted British attitudes. During late 1942 and early 1943, almost 90% of British people supported and approved the USSR (Ward 2012, p. 506). The years of the Anglo–Soviet Alliance (1941–1945) became a time of exceptional enthusiasm for supporting Russians, and the phrases "We pledge all our aid to Russia" and "All aid for the Soviet" from Prime Minister Winston Churchill in his 22 June 1941 radio broadcast were regarded as a call to action by thousands of British people of every society level (Knight 2012, p. 253). The British people collected 7.5 million pounds to help the USSR. Mrs. Clementine Churchill personally supervised these donations. Parcels with medical equipment and clothing continued to be transported to Russia until 1950. Hundreds of letters were sent to the Soviet Embassy in London and SCR. The Society received new branches in the cities of England and Scotland, and there were famous writers, artists, and film actors among those members (including Vivien Leigh and Sir Laurence Olivier, Somerset Maugham,

and Stephen Spender). In Scotland, the "Russia Today" society was organized to collect signatures expressing admiration for the courage of Leningrad residents (in the Summer of 1942, artists Anna Ostroumova-Lebedeva and Vera Milyutina prepared a response gift from Leningrad women—an album with lithographs (Henderson 1988)). Moscow women sent 13 notebooks with greetings to the wives of mayors of 13 London boroughs. SCR, together with the Women's Anglo–Soviet Committee, initiated the movement "Greetings from English schools to Soviet schools." Local Anglo–Soviet friendship weeks were held throughout the country (SCR 1942; SCR 1946a, 1946b).

During the 1940s, the SCR organized dozens of events—lectures, exhibitions, concerts, and festivals of Soviet cinema. The society published the *Anglo–Soviet Journal* quarterly. In 1942 The Royal Albert Hall hosted a concert in support of the collection of medicines for Russia (conductor Malcolm Sargent). Dmitri Shostakovich's Symphony No. 7 ("Leningrad Symphony") was performed by the London Philharmonic Orchestra (conductor Sir Henry Joseph Wood). On 27 June 1943, a rally and a concert were held in the Albert Hall in memory of the second anniversary of the Fascist invasion of the Soviet Union. In 1942, in memory of the anniversary of the invasion of the Fascist troops in the USSR, the SCR organized a traveling "Eastern Front Exhibition", designed by architect Erno Goldfinger.[2] The show displayed military ammunition, pictures from the front, and stands about Anglo–Soviet friendship. Then, in 1943–1944 the other exhibitions followed: two expositions about the beginning of the reconstruction in the Soviet Union and the exhibition "Hero Cities: Leningrad and Stalingrad" (at "Harrods," then at Whitechapel Gallery). At the peak of SCR activity, up to 7000 visitors came to the exhibitions. So, the exposition in 1944 was viewed by 7250 people (and this is only in the Whitechapel Gallery, then the exhibition still traveled to other venues) (Lygo 2013).[3] Even the Minister of Information, Member of Parliament, conservative Brendan Bracken took part in these events, opening an exhibition on the Eastern Front together with the USSR Ambassador Ivan Maisky.

Many recent emigrants tried to find a job in the Ministry of Information or any other department, using their knowledge of Russian and familiarity with the culture and nature of the territories that were parts of the former Russian Empire. Architect Berthold Lubetkin wrote to his friend that he wanted to be closer to the front, that he "would like to get a job with one of the Services if possible where either my architectural and general organizational qualifications could be made use of, or my linguistic qualifications, and knowledge I have of Southern Russia and my native Caucasus < . . . > The ideal job, I imagine, would be somewhere in the Middle East."[4] He was interviewed in 1942 in many establishments and wrote to various departments, but to no avail. He corresponded with The King's Own Royal Regiment and forwarded them some recollections about the Caspian Sea, Turkmenia, and Karakoum[5]. His friend from the Royal Air Force said that "there seems to be a complete 'stop' at any new appointments with the Russians, except those made on a level which does not come within my province < . . . > There is a recent Air Ministry Order asking for names of people who can speak odd languages including those which you speak, but these jobs are all purely clerk or regular interpreter classes < . . . > The other point I have gathered is that the Russians themselves prefer non-Russian-born personnel on the Liaison staff. I believe, in your case, this is one of the principal snags. However, why this should be so, at this stage of the war, I cannot tell."[6] Perhaps that is why Lubetkin, in one of the questionnaires for the Royal Air Force Volunteer Reserve, indicated his European origin and named Warsaw as his place of birth.[7]

On the one hand, the Ministry of Information popularized the heroism of Soviet soldiers but, at the same time, monitored pro-Soviet organizations and sought to minimize their influence (Lygo 2015). The enthusiasm for communism was not shared by everyone, and naturally, it did not cause encouragement from the British authorities. So, in 1942, architect Berthold Lubetkin designed and initiated the installation of a commemorative bust of Lenin in Holford Square (where the politician lived between 1902 and 1903). The bust did not stand for long, causing outrage among local residents. As a result, Lubetkin

and his colleagues dismantled the monument with a crane and "buried" it right on the square (Allan 1992).

## 5. Professional Correspondence and Discussion of Vital Architectural Questions

Under all these conditions, architecture and urban planning undoubtedly became one of the main subjects of discussion. One of the first messages with words of support to the Union of Soviet Architects was received via VOX from William Henry Ansell, President of RIBA, on 2 July 1941, promising that British architects would be united with Soviet colleagues[8]. In response, the Soviet architects sent a radiogram assuring the following: "We architects have placed ourselves at the disposal of our government, which leads the people to battle against the rapacious aggressor stop < . . . > In this historic hour, we express our deep friendship for our British colleagues and the people of Britain."[9] "Kind regards in these difficult times"—these words of encouragement from Harold Elorn, a British architect, who wanted to work in Moscow, is one of the examples of many letters sent to VOKS in 1941.[10] The Union of Soviet Architects also received letters from British students who wrote that they wanted to come to work in Russia after the war, but in the meantime, they dreamed of finding a Russian friend, an architect student, by correspondence. The Association of Architects, Surveyors, and Technical Assistants (AASTA, a professional trade union with strong Communist involvement) also sent a radiogram expressing a strong desire to move together to victory and post-war reconstruction.[11] Even Sir Patrick Abercrombie, thanking Soviet architects for the calendar, in August 1944 said that his son was "an ardent admirer of all things Russian and looks to visiting Russia after the war."[12]

In August 1942, the *Journal of the RIBA* quoted architect Victor Vesnin's radio speech confirming that while there was a struggle, friendship with England was growing stronger.[13] In this radio report Vesnin, who was later awarded a gold medal of RIBA in 1944, mentioned the contribution of English architects to urban planning and the legacy of British authors in Russian architecture. He also broached the most significant topics that were then relevant for the USSR and Britain: the importance of sharing experience in matters of air defense and the development of projects for the restoration of destroyed cities.[14] Before the war, the experience of Soviet architecture attracted the attention of British specialists since, in the 1920s and 1930s, most European cities were at a crossroads, deciding how to develop a modern city. Those who managed to visit Russia (Ernest Simon, Williams-Ellis, Berthold Lubetkin, Arthur Ling, Jaqueline Tyrwhitt) talked about the experience of the USSR (Lubetkin 1933; Ling 1941, 1942; Shoshkes 2006). SCR distributed the monthly Soviet propaganda magazine *The USSR in Construction*. In 1937, the book *Moscow in the Making* was published, which was popular, especially among pro-communist English architects, who thought that more progressive achievements were possible in Russia than in conservative Britain (Simon et al. 1937; Simon 1945). Urban planning issues turned into a problem of vital importance after the bombing of English cities, so logically, communication on these topics was strongly encouraged. The large-scale destruction of architectural monuments, as well as the task of replenishing the lost housing stock in a short time, the problems of both the USSR and England, contributed to the interest in the exchange of experience.

The Society for Cultural Relations of the USSR—Great Britain established through VOKS an exchange of architectural periodicals and books on current professional topics; British architects regularly (as far as it was possible in wartime conditions) received the magazine *Architecture of the USSR*. In 1944, the SCR launched a duplicate bulletin, *Soviet Reconstruction Series: Town-planning, architecture, and building*. These brochures were devoted to plans for the restoration of cities and projects for constructing new districts. The requests for books on the history of Russian architecture and about modern construction and materials were sent to VOKS from The Incorporated Association of Architects and Surveyors, Association for Planning and Regional Reconstruction, British Commercial Gas Association (written personally by Jane Drew), Town and Country Planning Association (request for photos of the soviet buildings for the *Design in Civil Architecture* edition), British Standards Institution, National Buildings Record, Department of Scientific and Industrial

Research. The number of technical questions was endless: modular system, dimensional coordination, effects on the stability of buildings' foundations during freezing and thawing, new methods of protecting freshly-laid concrete from frost, use of mechanical aids on the building site, movable scaffolding or shuttering, cranes or transporters, peat-cement mixtures, and the technical-economical basis of the settlements planning. Having such an increased interest, SCR invited Alabyan to exchange documentaries on the progress of the restoration of Stalingrad and other cities.

## 6. The Union of Soviet Architects and Its International Interests

This communication was no less important and necessary for Soviet architects. In fact, the Architectural Section of VOKS started sending the *Architecture of the USSR* magazine to SCR and RIBA in 1942, amid the battle for Moscow, when Hitler's troops were close to the capital. Next year, the Member of the VOKS Board, Lydiya Kislova, corresponded with Sir Albert Edward Richardson, asking for advice about the best ways of collaborating with British and Soviet architects.[15] According to the notes on the draft of the letter, the same requests were intended to be sent to Sir Edwin Lutyens, William Henry Ansell, Sir Jan MacAlister, William Graham Holford, Sir Banister Fletcher, Prof. Stanley Davenport, Sir Clough Williams Ellis, Mr. D. Percival, Rober Atkinson, Sir Percy E. Thomas, and Sir Patrick Abercrombie. VOKS tried to use all the possibilities of postal communication, observing correspondence etiquette, despite wartime. In 1944, members of the Architectural Section were scolded by VOKS for responding slowly and untimely to letters from British colleagues. They sent all the necessary congratulations and condolences (for instance, to Sir Edwin Landseer Lutyen's relatives). Since RIBA chose architects Karo Alabyan, Nikolai Kolli, Alexander Nikolsky, David Arkin, and Grigory Simonov as their members in 1936, the Union of Soviet Architects considered it was also necessary to select in response W.H. Ansell, Sir Banister Fletcher, Sir Jan MacAlister, Sir Percy E. Thomas as its honorary members.

The Union of Soviet Architects secretary informed the Agitation and Propaganda Department of the Central Committee of the Communist Party of the Soviet Union that in connection with the conclusion of the Anglo–Soviet Treaty of 1942 and the first anniversary of the war, they were carrying out a number of activities to strengthen cultural ties with the architectural community of Great Britain. In February 1942, Karo Alabyan asked VOKS to help with materials about the British Reconstruction experience and accelerated wartime construction as the Academy of Architecture of USSR started working on the same questions. Alabyan even ordered some texts in particular: John Reith's report from 26 February 1941 and the Royal Commission's report on the geographical distribution of the Industrial Population.[16] According to the letters sent via VOKS by the Soviet architects, a lot of current materials were received from London. Sir Percy E. Thomas sent the *Greater London Plan*, Charles Reily — the edition of *The Reilly Plan — a New Way of Life*. Alabyan received several editions like *Town and Country Planning* by Patrick Abercrombie, *New Town after the War* by F.J. Osborn, *The Design of Nursery and Elementary Schools* by R. Gardner-Medwin, H. Myles-Wright, *Housing Before the War and After*. M.J. Elsas, *Reconstruction and Town and Country Planning*. Sir Gwilym Gibbon, *The Bombed Buildings of Britain* by Ed. G.M. Richards.[17] Professor N. Vetchinkin (head of Sector of Civil Engineering Academy of Municipal Economy) corresponded with Sir Charles Reilly, asking about pre-fabricated housing and industrialization of building operations. He asked for assistance collecting "technical information on the repair and restoration of residential buildings damaged or destroyed as a result of German air raids."[18]

In 1944, the British city Coventry, ruined by air raids, began its Bond of Friendship relationship with Stalingrad. It was the time when the National Council of British–Soviet Unity initiated Fraternal Relations Between British and Soviet Towns. It was supposed to exchange greetings, souvenirs, and information between urban workers, engineers, and medical staff. In 1945, vice-president of the National Council sent a project of Memorandum to VOKS, describing the situation rather emotionally: "In Britain the Society for Cultural Relations (S.C.R.) has done, and is doing very valuable work. But it is not a

mass organization. It represents chiefly the intelligentsia and artistic and cultural circles. The organization which mobilizes mass support is the National Council for British-Soviet Friendship, has 300 Committees throughout Britain. These Committees are on a broad basis and usually represent all political parties and religious, cultural and trade union groups < . . . > This is a new phenomenon and may not last now that the war is over unless measures are taken to hold the interest created. < . . . > With the ending of the war and release from Nazis of countries like Holland and Norway, unless the closest relations are maintained and sustained by every means, assistance, and sympathy may be diverted from the U.S.S.R." Since not all mayors responded to the offer of friendship, it was proposed to build relations only with those cities that quickly and willingly desired to establish contacts. The National Council asked to correspond only via VOKS and respond as quickly as possible to consolidate the unprecedented enthusiasm.

In 1945, the Architecture and Planning Group was formed in SCR, which included well-known British architects (for example, Sir Charles Reilly, Vice-President of the group), Sir Patrick Abercrombie, MARS group Wells Coates representatives, Arthur Ling, and Berthold Lubetkin. Many of them played an essential role in British urban development, preparing the 1943 County of London Plan (Ling) and taking responsibility for many new post-war dwelling districts. About 60 architects attended the first lecture, organized by the new Planning group. The head of VOKS Architectural Section, Karo Alabyan, (during the war, he was leading the work of the Union of Soviet Architects and the Academy of Architecture) heartily congratulated British colleagues, noticing that destructions caused by war confronts architects of Great Britain and the USSR with stupendous tasks of the post-war reconstruction. Karo Alabyan and David Arkin began to exchange information intensively. Arthur Ling, "anxious to explore," requested from Soviet architects: reports about the organization of the architectural practice in the USSR, information about preferred styles ("realism and the creation of people's architecture"), the influence of new materials and methods of construction, a description of types of flats and houses including pre-fabricated types. In November 1945, Arkin wrote to Ling: "Keep in mind that we are especially interested in modern housing construction (including the internal equipment of buildings), as well as the layout of settlements, landscaping."[19]

As part of the program for cultural cooperation with England, the Union of Soviet Architects Board held "English days" dedicated to British architecture. During those meetings, the architects became acquainted with books and photographs sent from the RIBA and gave lectures on classical and modern British architecture. The Architecture Section also organized radio programs about Anglo–Soviet ties. It was decided to show the materials of photo exhibitions sent from Britain in 1944 (on the construction of low-rise buildings and about the new plan for London) as a large exhibition in the House of Architects. Discussing the photos, the section members regretted that it was impossible to send back similar images of Russian architecture due to the lack of photo paper. Academician Alexey Shchusev lamented: "The bottleneck is photo paper!"[20] In the absence of good photos, the VOKS sent books and articles on historical architecture. VOKS also attached to them lists of the monuments destroyed during the war.

A remarkable fact, in the second half of the 1930s, many texts and photographs of foreign architectural projects were regularly published in the official Soviet architectural periodicals. There were historical reviews and analyses of modern practice. But mostly, the practice of France, Italy, the Netherlands, and America was covered. English architecture was rarely mentioned, and more often when it came to building and engineering news, city planning, and anti-aircraft camouflage. During the war, on the contrary, almost the only topic in the review of foreign experience becomes the British and American experience, mainly residential dwelling and urban redevelopment.

## 7. The Problem of the Lost Heritage and Post-War Restoration

At the same time, forwarding exhibitions about Russian and Soviet architecture to London and other English cities were urgently needed. Even in 1942, at one of the meetings

of the Union of Soviet Architects Board, this need was already admitted by Russian Architecture Propaganda Section: "The main task of the Section is to promote the architectural culture of the USSR and its member countries in the past and present, by organizing exhibitions, giving lectures and reports on Russian architecture, the architecture of the peoples of the USSR and Soviet architecture and art, organizing radio appearances, publishing articles in general and special Russian and foreign magazines, making documentaries, producing postcards and a special bulletin dedicated to the monuments destroyed by the German fascists in the liberated."[21] It was offered to show 250–300 exhibits in the foyers of theaters, cinemas, clubs, military units, and public institutions.

British and Soviet architects saw the problem of scientific restoration as no less important than the restoration of cities as a whole and the need to replenish the housing stock with the help of serial dwellings and prefabrication (Bullock and Verpoest 2011). The scale of the historical monuments' destruction was the most painful and frequently discussed in professional correspondence. The British, who also lost valuable architectural and historical heritage as a result of the bombing, could share and understand the emotions of their Soviet colleagues. The analysis of the texts allows us to say that the letters on this topic were the least formal. After greeting Sir Percy E. Thomas in his message, David Arkin wrote that he had read Percy's cable in the newspaper "with great satisfaction" (24 September 1945). The British architect declared indignation at the destruction of outstanding monuments: "We know that all of you in Britain have also lived through a bitter loss of remarkable monuments of British architecture < . . . > Feelings expressed by you on behalf of our British colleagues will find an immediate response not only among Russian architects but among all those who appreciate the treasures of human culture and immortal creations of art." Arkin listed numerous destroyed unique medieval Russian monuments and stressed that cathedrals of London, Coventry, Exeter, and Canterbury were "dear to us as treasures of universal human value."[22] The past also became a subject of discussion with A.E. Richardson, who wrote to VOKS in 1945: "First let me condole with my Russian Architect Colleagues on the wanton damage committed by the German vandals. We in England know what it means to lose priceless masterpieces of Art < . . . > I see in the fine organization of Architectural research which the Soviet Government has set in motion the basis of a great school of Russian Architecture. Russia has wonderful and varied traditions that have crystalized throughout the centuries in her buildings < . . . > The publications sent to us make these and other facts clear."[23] In the correspondence between architects, in particular, the common plot of Russian and English XVIII century neoclassicism is considered. Each party saw one of the development peaks of national art in these works. This topic was discussed in detail by David Arkin and Richardson, who emphasized that outstanding national examples appeared thanks to the international, Pan-European XVIII century classicism. Richardson admired old Saint-Petersburg and Moscow architecture. At that time, he was working on the publication dedicated, as he told Arkin, to "classical architecture as belonging to all nations."[24]

The classical heritage of Russian architecture was discussed in correspondence between VOKS and The Georgian Group. The *Studio* and the *Plan* magazines published articles on the history of Russian architecture by famous historians—Viktor Lazarev and Igor Grabar. It became possible to transfer books from Russia to Britain from historian Vladislav Lukomski to his brother George Loukomski—a critic and art historian who emigrated to Western Europe after 1917. To make this handover possible, the Soviet Embassy in London prepared a report for VOKS about historian George Loukomski in 1943, saying that he continued writing about Russian architecture and Russian painting and that he owned a good collection of works by Russian artists, but "How it treats us is unclear."[25]

After the successful photo exhibition of destroyed Russian cities in England in 1944, there were lengthy negotiations on organizing a larger-scale display. At a meeting of the Architectural Section, L. Kislova said: "Despite its modest size, this exhibition made a huge impression. This can be judged by the English press . . . the nature of the destruction caused deep feelings, and many got acquainted with the monuments that they still did not



know through the pictures of the destroyed buildings.- Pictures of Volokolamsk Monastery have bypassed several magazines." Kislova suggested publishing the photos "before and after", in a way she saw in the book about England composed by British colleagues. The preparation of the new exhibition was slow; there was no opportunity to photograph the destroyed monuments in the newly liberated territories. As a result, an exhibition was prepared after the war: "Exhibition of the Architecture of the USSR," opened at RIBA by the Soviet Ambassador and Sir Lancelot Keay in March 1948. The show was attended by 4000 people over 17 days.

Developing the concept, the authors proposed three parts: a historical essay, a section about modern Soviet architecture, and the restoration work after the war (telling how "the Soviet epoch gave architecture back to the people"). The emphasis was made on new typologies and industrial cities. The exposition was supposed to show a unique experiment and the opportunities offered by the absence of the private sector, "not only as a purely technical project but also as an architectural ensemble of the highest artistic rank." While preparing the explications, Russian curators had already understood that the question of the style would be widely discussed, especially by the audience who are skeptical of Soviet practice, so they explained as follows: "Natural ornamental marbles of various kinds were employed not as a deliberate attempt at "luxury" but as the most expedient manner of utilizing materials in which various districts of the country are rich, materials which can be put to the best artistic and technical use precisely in structures of this kind."[26] These remarks already indicated a change in the assessment of mutual experience.

## 8. Conclusions

When in May 1945, Mrs. Churchill received a telegram from Winston Churchill begging her to return to London, she refused and extended her visit to USSR (Knight 2012, p. 261). Unless the political tensions had come back by inertia, some earlier negotiations continued from 1945 to 1947 (like an exhibition of contemporary British graphic art in Moscow or a series of three volumes, "The Soviets and Ourselves", published with the aim "to promote understanding and prevent misunderstanding < . . . > to understand is to recognize unity in difference") (Macmurray 1945, p. 5). Gradually the Anglo–Soviet relations became strained (Van Oudenaren 1991; Atkinson and Verity 2017). Right after the victory, the fear of public enthusiasm for communism altered the attitudes again, escalating into the beginning of the Cold War.[27] In response to criticism of Soviet graphics shown in April 1945 at the Royal Academy of Arts, one Russian critic noted: "In general, our English friends have an idea about us, I would say, so far friendly, but somewhat strange."[28] Letters from the UK to VOKS slowly stopped. The anti-Western agenda appeared in the Soviet architectural periodicals (Vlasov 1948).

When the Soviet Committee for Architectural Affairs took the reins in 1943, the nature and tone of the controversy changed. The head of the Agency for Planning and Developing Cities at the Committee for Architecture, Viktor Baburov, stated that new technologies and standardization prevented the West from building decent monuments. Baburov said that Western architects (in Birmingham and Chicago) did not seem to care about the problem of artistic urban planning. He agreed that in matters of comfort, the English architects had achieved fantastic results, while the Soviets put forward monumental tasks of urban planning instead (Chmelnizki 2013, p. 328; Kosenkova 2000, p. 219; Kosenkova 2010). Reconstruction in the Soviet bloc took the shape of socialist realism (later turned to socialist modernism), which caused a kind of "socialist globalization" (Czarnecki and Chodorowski 2021, p. 4).

Even in spite of these official changes, Soviet architects regularly received invitations to come to England right after the war. Unlike before the war, when the Soviet government stimulated professional architectural contacts, during the war, no orders were officially issued on the need to exchange architectural experience. On the contrary, no matter how many times SCR wrote to Moscow that an exchange of delegations was needed or Karo Alabyan asked to send a group of Soviet architects to Britain, nothing worked.

The Chairman of the Committee for Architecture under the Council of Ministers of the USSR, Grigory Simonov, explained that due to the extensive restoration work, Soviet architects could not take part in the trips to London (Malich 2021, p. 785). Only in the 1950s the Russian architects finally went on business trips abroad: in 1952, they made trips to Romania and the Czech Republic; in 1953, delegations were sent to Portugal, Finland, Romania, France, East Germany, and Poland. Foreign colleagues came to Moscow on a return visit. In September 1953, the first group of British architects made a three-week journey to the USSR at the invitation of the Union of Architects of the USSR, providing a chance for the next "Indian summer" in Anglo–Soviet relations.

Following the questions we asked and defining the criteria for evaluating professional contacts, we can conclude that there were no officially declared intentions about the need for professional exchange. The architects did not communicate within the framework of any governmental program but at the semi-official level policy. All the correspondence was conducted through the Union of Soviet Architects. But it was the architects themselves who formulated the request for communication with British colleagues. It should also be noted that before the war, *The Architect of the USSR* magazine constantly published reports on French, Italian, and Dutch architecture, then, during the war, if we are talking about foreign practice, then mainly materials about England and the USA are published. In the case of British architects, communication was primarily initiated by SCR, although there were also some private initiatives.

This exchange of experience proved to be the most valuable in the field of restoration of destroyed cities. However, it was extremely difficult to put foreign experience into practice; the conditions of the construction infrastructure were too different. To apply foreign methods, it was necessary to have closer communication, visual training, and detailed drawings. Both British and Soviet architects were interested in the new town planning concepts and tried to find solutions to reform the building industry. Sympathy for the hardships of the war and the loss of historical monuments was sincere in both countries, prompting to exchange letters, books, and exhibitions. For the Soviet architects, it was a rare moment when the reason for corresponding was not the necessity to prove the advantages of the Soviet system but to become aware of the absolute need to restore monuments and raise the construction industry. It was a short but exceptional period that could have left a much more significant mark and resulted in a more practical way. In fact, under the pressure of political circumstances, it was almost unreal to implement decisions that were elaborated in different economic situations with different management and social systems.

**Funding:** This research received no external funding.

**Data Availability Statement:** Not applicable.

**Conflicts of Interest:** The author declares no conflict of interest.

## Notes

[1] GNIMA archive, fund No 49, inventory 3, item 167, pp. 1–5.
[2] RIBA Archive AP 66/65.
[3] SCR Annual Report 1944, SCR Archive.
[4] RIBA Archive LuB/11/2/28.
[5] RIBA Archive LuB/11/2/17.
[6] RIBA Archive LuB/11/2/22.
[7] RIBA Archive LuB/11/2/37.
[8] RGALI archive, fund 674, inventory 2, item 85, p. 2.
[9] RGALI archive, fund 674, inventory 2, item 85, p. 3.
[10] RGALI archive, fund 674, inventory 2, item 85, p. 1.
[11] RGALI archive, fund 674, inventory 2, item 99, p. 62.
[12] GARF archive, fund 5283, inventory 1, item 197, p. 10.

13    Anglo–Soviet Bonds in Architecture. *Journal of the RIBA*. August 1942. p. 164.

14    RGALI archive, fund 674, inventory 2, item 99, p. 38–40.

15    GARF archive, fund 5283, inventory 15, item 142, p. 35.

16    GARF archive, fund 5283, inventory 15, item 198, p. 34.

17    GARF archive, fund 5283, inventory 15, item 197, p.23–24.

18    GARF archive, fund 5283, inventory 15, item 282, p. 1, 7–9, 14.

19    GARF archive, fund 5283, inventory 15, item 282, p. 2.

20    GARF archive, fund 5283, inventory 21, item 23, p. 18.

21    RGALI archive, fund 674, inventory 2, item 99, p. 108.

22    GARF archive, fund 5283, inventory 15, item 291, p. 31.

23    GARF archive, fund 5283, inventory 15, item 142, p. 12.

24    GARF archive, fund 5283, inventory 15, item 142, p. 1.

25    GARF archive, fund 5283, inventory 15, item 198, p. 28–29.

26    GARF archive, fund 5283, inventory 15, item 167, p. 5–52.

27    Soviet Writers Reply to English Writers' Questions. Ed. by Edgell Rickword. London: The Writers' Group, Society for Cultural Relations with the USSR, 1948. p. 5.

28    GARF archive, fund 5283, inventory 15, item 167, p. 151–54.

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
