# Peer review of "Kind Regards in These Difficult Times: Anglo–Soviet Architectural Relations during the Second World War"

_arts, 2023_

Round 1

Reviewer 1 Report

This manuscript lacks original findings and interpretations. It mostly just repeats facts collected from already published sources. In addition, you should add opinions, interpretations, and more detailed reviews about the various historical events and exhibitions you mention.

In addition to quoting letters and describing activities, you should also add more mention of architecture (designs, reviews, critiques, etc.). Did anything come out of these contacts and exchanges between English and Russian architects? You should also describe what was shown in the exhibitions you mention.

In particular, I did not find much mention of the two points you list in the introduction: Human factor and the difference between ideological and non-ideological practices and ideas.

I also cannot make sense of your footnotes.

Author Response

Dear colleagues, thank you for your comments. I'll try to make my response convincing.

1). This manuscript lacks original findings and interpretations. It mostly just repeats facts collected from already published sources. In addition, you should add opinions, interpretations, and more detailed reviews about the various historical events and exhibitions you mention.

The relations between Soviet and British architects during the II World War were never observed by the historians. All the facts are presented for the first time and are based on the archive materials that were never published before. The only exception - is the review of general relationships between UK and USSR in the paragraph 2.

2) In addition to quoting letters and describing activities, you should also add more mention of architecture (designs, reviews, critiques, etc.). Did anything come out of these contacts and exchanges between English and Russian architects? You should also describe what was shown in the exhibitions you mention.

I would add the conclusion concerning this moment, thank you! To sum up, as I see, it was of cause more about the intentions than about the real practical architectural exchange. 

3) In particular, I did not find much mention of the two points you list in the introduction: Human factor and the difference between ideological and non-ideological practices and ideas.

I will try to make it more clear text, adding some comments. Sympathy for the hardships of the war and the loss of historical monuments was sincere, prompted many British colleagues to write to Soviet architects. And for the Soviet architects it was a rare moment, when the reason to correspond wasn't that it was necessary to prove the advantages of the Soviet system, but in the real needs to restore monuments and raising construction industry.

4) I also cannot make sense of your footnotes.

Usually I refer to archives, putting the name of the archive, the number of archive collection, folder and paper. Archive of the Russian Federation (GARF), the Russian State Archive of Literature and Art (RGALI), the Royal Institute of British Architects (RIBA).

Reviewer 2 Report

This paper titled “-Kind Regard in these difficult times-. Anglo-Soviet architectural relations during the Second World War” presents an informative overview of the architectural relationships between the United Kingdom and the Soviet Union in the 1940s.

The paper showcases interesting details and examples; and it demonstrates thorough archival research. However, the text does not follow a research paper’s structure. It would be crucial to reorganise all the information following a clearer and more effective format for the article. I would recommend authors follow the journal’s guidelines regarding paragraphs.

The paper needs a proper introduction, conclusion, and more effective development. What is the research question? What is the thesis? What are the methods used, and what are the findings?

There needs to be an explanation of the main aims, a description of the research methods, and a discussion of the results.

Furthermore, the use of standard English could be improved. It would be beneficial to avoid contracted verb forms.

Finally, it would be essential to include more references. Understandably, not many papers have been published focusing on the architectural relations between the UK and the Soviet Union during the Second World War. However, it would be important to refer to research on each country’s architectural situation during the same period of history.

Author Response

Dear colleagues, thank you so much for your time and comments. I'll try to reorganize the article and add a proper introduction, reasoning, conclusion,  and references. I do hope it will help to make the text better and more convincing!

Kind reagards, yours, Ksenia Malich

Round 2

Reviewer 1 Report

The article is marginally better than the previous version. It is just purely factual, and hence a bit boring to read.

There is an interesting background to your research. At the beginning of the 20th century, Soviet architecture was at the forefront of Modernism. After the fall of the iron curtain, Western architects were involved for a short period in updating Russian architecture. Today, there is obviously deep antipathy between Russia and the West. You could position your focus on exchanges during WWII into this broad development.

Author Response

Dear colleagues, I am really thankful for your time and reviews. At the beginning of the 20th century, Soviet architecture was indeed at the forefront of Modernism. But, as I noticed in the research, the British architects were very skeptical about these changes. The situation become even worse during the second part of the 1930-s. During the war everything changed. I tried to explain all points more clear in the Introduction. I tried to present unpublished documents, showing intensive exchanges between British and Soviet colleagues, because of the common problems they had to solve giving both sides the hope for future collaboration and postponing ideological confrontation. As you see, it was really very emotional moment. But unfortunately the beginning of the Cold War and different building infrastructure opportunities prevented architects  from further productive cooperation. The paths dramatically diverged. 

Thank you for you help and comments. I will continue improving trying to do my best.

Warm regards

- Removed for Peer-review -

Reviewer 2 Report

Dear Author(s),

Thank you for revising the manuscript.

The references are now more comprehensive. Also, the introduction has been improved.

However, there is still no explanation of the research methods, and the structure is still unclear. Therefore, I would suggest the author(s) to follow the journal's guidelines and be more effective in explaining the methodology applied to conduct the study, the main aims, significance and relevance of the study (one sentence in the introduction is not sufficient), and finally to rewrite the conclusions carefully.

Best regards

Author Response

Dear Sirs, 

Thank you for you help and comments. I have examined a lot of recent articles from "Arts" trying to follow and repeat exactly the common guidelines. I tried to review Introduction and Conclusion to make them more logical and hope now it will be more consistent.

My warm regards,

- Removed for Peer-review -

Round 3

Reviewer 2 Report

Overall, the revisions made by the author have significantly improved the clarity and coherence of the paper. The author has addressed several issues raised in the previous review. However, there are still three key areas that require further attention.

Firstly, the paper lacks a clear statement regarding the methodology employed for the research. The author should provide an overview of the data collection process and analysis techniques.

Secondly, the paper would benefit from a clearer identification of the main aims of the research. Although the author has mentioned the objectives briefly, they are not explicitly stated and may be overlooked by readers. By clearly defining the research aims, the author can effectively guide the readers' understanding and provide a framework for the subsequent sections of the paper. This will help readers grasp the purpose and significance of the study from the outset.

Lastly, the paper requires proofreading to address minor mistakes in English. A thorough proofreading session would greatly enhance the clarity and precision of the language used throughout the manuscript.

Author Response

Dear Sirs, I have added some words about the methodology and the main aims. I have also looked through the paper with my English speaking colleague. Thank you again and again for your time, help and care. 

Yours, -Name removed for peer-review-